# Mycoviruses Increase the Attractiveness of *Fusarium graminearum* for Fungivores and Suppress Production of the Mycotoxin Deoxynivalenol

**DOI:** 10.3390/toxins16030131

**Published:** 2024-03-02

**Authors:** Simon Schiwek, Matthäus Slonka, Mohammad Alhussein, Dennis Knierim, Paolo Margaria, Hanna Rose, Katja R. Richert-Pöggeler, Michael Rostás, Petr Karlovsky

**Affiliations:** 1Institute for Plant Protection in Field Crops and Grassland, Julius Kühn Institute (JKI)—Federal Research Centre for Cultivated Plants, 38104 Braunschweig, Germany; 2Molecular Phytopathology and Mycotoxin Research, University of Göttingen, 37077 Göttingen, Germany; 3Agricultural Entomology, University of Göttingen, 37077 Göttingen, Germany; matthaeus.slonka@uni-goettingen.de (M.S.);; 4Leibniz Institute DSMZ-German Culture Collection for Microorganisms and Cell Cultures, 38124 Brunswick, Germany; dennis.knierim@dsmz.de (D.K.); paolo.margaria@dsmz.de (P.M.); 5Institute of Horticultural Production Systems, University of Hannover, 30419 Hannover, Germany; 6Institute for Epidemiology and Pathogen Diagnostics, Julius Kühn Institute (JKI)—Federal Research Centre for Cultivated Plants, 38104 Braunschweig, Germany

**Keywords:** mycovirus, *Fusarium graminearum*, trichothecenes, deoxynivalenol, aurofusarin, VOC, Collembola, fungivory, food preference, *Folsomia candida*

## Abstract

RNA viruses of the genera *Ambivirus*, *Mitovirus*, *Sclerotimonavirus*, and *Partitivirus* were found in a single isolate of *Fusarium graminearum*. The genomes of the mitovirus, sclerotimonavirus, and partitivirus were assigned to previously described viruses, whereas the ambivirus genome putatively represents a new species, named *Fusarium graminearum ambivirus* 1 (FgAV1). To investigate the effect of mycoviruses on the fungal phenotype, the spontaneous loss of mycoviruses during meiosis and the transmission of mycoviruses into a new strain via anastomosis were used to obtain isogenic *F. graminearum* strains both with and without mycoviruses. Notable effects observed in mycovirus-harboring strains were (i) the suppression of the synthesis of trichothecene mycotoxins and their precursor trichodiene, (ii) the suppression of the synthesis of the defense compound aurofusarin, (iii) the stimulation of the emission of 2-methyl-1-butanol and 3-methyl-1-butanol, and (iv) the increased attractiveness of fungal mycelia for fungivorous collembolans. The increased attractiveness of mycovirus-infected filamentous fungi to animal predators opens new perspectives on the ecological implications of the infection of fungi with viruses.

## 1. Introduction

Gibberella ear rot, caused by the pathogen *Fusarium graminearum*, is a widely spread maize disease causing severe yield losses and grain quality decline [1]. The contamination of grain with mycotoxins of *Fusarium* spp. threatens food and feed safety [2,3,4]. Ears infected with *F. graminearum* often contain deoxynivalenol (DON) [5] or nivalenol (NIV) [6] and their acetylated derivatives, which act as virulence factors [7] by suppressing plant defense responses [8]. Furthermore, the pathogen produces the estrogenic metabolite zearalenone [9], which protects the fungus against mycoparasites [10], and the bis-naphthopyrone pigment aurofusarin [11], which protects the fungus against predators [12].

Fungi-infecting viruses (mycoviruses) have been described in most fungal taxa [13,14]. The genome of mycoviruses typically consists of single-stranded or double-stranded RNA; genomes consisting of DNA are less common. Despite a high level of genomic and structural diversity, most viruses can be taxonomically classified based on the conserved motifs of their RNA-dependent RNA polymerase (RdRp) [15]. Mycoviruses are transmitted by cell fusion (hyphal anastomosis) or sporulation. The dissemination of mycoviruses by fungivores has only been documented in a single case so far [16].

The most extensively studied effect of mycoviruses on fungal phenotype is the suppression of aggressiveness in plant pathogens, termed hypovirulence [17,18], which gained attention due to its potential for the biological control of plant diseases [19,20]. *F. graminearum* is an economically important pathogen and producer of mycotoxins with a significant impact on food safety. Examples of mycoviruses inducing hypovirulence in *F. graminearum* are the dsRNA virus FgV1 [21], *F. gramineaum virus china-9* (FgV-ch9) [22], and *F. graminearum hypovirus 2* (FgHV2) [23]. The cause of the suppression of aggressiveness in *F. graminearum* by mycoviruses is unknown. Studies in fungi other than *Fusarium* spp. showed that mycoviruses may also increase the aggressiveness of plant pathogens, which is called hypervirulence [14,24].

So far, few studies have investigated the effect of mycoviruses on mycotoxin production. The first mycovirus found in *F. graminearum* was reported to cause a 60-fold decrease in the production of trichothecenes [21]. The claim was perpetuated in the literature, though the “60-fold decrease” was due to a calculation error (see Section 3.4), and the suppression of trichothecene synthesis was not proven. Twelve years later, the same laboratory showed that FgV1 suppressed the agmatine-induced synthesis of trichothecenes in *Fusarium gramineaum*, while three other mycoviruses did not substantially affect trichothecene synthesis [25]. On the contrary, two studies reported the stimulation of mycotoxin synthesis by mycoviruses. According to the first study, the transfection of *Aspergillus ochraceus* with a partitivirus from a different strain stimulated the production of ochratoxin A up to threefold [26]. Mycotoxin levels were normalized to fungal biomass, determined as the weight of mycelia harvested from liquid media. The second study reported that mycoviral infection triggered the synthesis of tenuazonic acid (TnA) in *Magnaporthe oryzae* [27]. The finding was surprising because *M. oryzae*, which is the causal agent of rice blast disease, was not known to produce TnA. The amount of TnA in fungal cultures was not normalized to fungal biomass, but the intensities of other HPLC signals in extracts of cultures both with and without the mycovirus were comparable, and TnA in strains without mycoviruses was undetectable, corroborating a causal relationship between mycotoxin infection and TnA production.

Except for a report on the dissemination of a mycovirus by a mycophagous predator [16], limited, however, to a single system and not yet independently reproduced, nothing is known about the effect of mycoviruses on the interactions of fungi with their predators. It is plausible to hypothesize that selection favors mycoviruses with the ability to render their hosts less attractive to predators, except for predators transmitting mycoviruses.

In this study, we report on an *F. graminearum* strain infected with four mycoviruses, we characterize a new member of the genus *ambivirus,* and we investigate the effect of mycoviruses on the radial growth of fungal colonies, the production of volatile and nonvolatile secondary metabolites, and the attractiveness of *F. graminearum* to the fungivorous arthropod *Folsomia candida*.

## 2. Results

### 2.1. Characterization of F. graminearum 266 MV Carrying Mycoviruses and Its Descendants That Lost Mycoviruses by Mitotic Segregation

Strain *F. graminearum* 266.1 was isolated from a maize ear from South Germany. The strain was putatively identified as a member of the *F. graminearum* species complex, based on the morphology of macroconidia and the deep red color of the colonies on agar plates. The taxonomic assignment was supported by the melting profiles of amplified fragments of genes encoding for the second largest subunit of RNA polymerase II (*RPB2*) and the translation elongation factor 1 alpha (*TEF-*1*α*) [28], and by sequencing a 503 bp fragment of the *TEF-*1*α* gene according to O’Donnell et al. [29]. The sequence was deposited in NCBI under accession number OP035498. The strain was later found to carry mycoviruses (see below) and, therefore, it was renamed as *F. graminearum* 266 MV.

Agarose electrophoresis of the total nucleic acids extracted from strain 266 MV revealed the presence of an extrachromosomal element of approximately 2.5 kb (Figure 1, lane 4). Treatment with DNase I (see Section 4.2 for details) completely removed the genomic DNA, but it left small RNAs and the 2.5 kb fragment unaffected, indicating that the 2.5 kb fragment consisted of RNA (Figure 1, lane 5). Note: no rRNA is visible (cf. to Figure 2) due to the incubation of samples in lanes 4 and 5 in Mg2+-containing buffer with (lane 5) or without (lane 4) DNase I. Magnesium ions in the buffer likely activated RNases, which are always present in unpurified nucleic acid extracts.

We hypothesized that the 2.5 kb RNA fragment originated from one or more mycoviruses. To generate a mycovirus-free strain and to investigate the effect of putative mycoviruses on the host, we attempted to cure the strain by treatments with cycloheximide and ribavirin; however, we had no success. Therefore, we grew strain 266 MV on PDA, purified about 200 single-spore isolates, extracted their nucleic acids, and screened the extracts using agarose electrophoresis for the loss of the extrachromosomal element. Indeed, one strain lost the extrachromosomal element visible in agarose electrophoresis (Figure 2C), and it was designated as *F. graminearum* 266.

On PDA plates, strain 266 MV exhibited an irregular growth pattern and dark red pigmentation, while strain 266 formed fluffy colonies with smooth margins. The color of the colonies of strain 266 were less intense, similar to most *F. graminearum* strains in our collection (Figure 2A,B).

### 2.2. Confirmation of the Presence of Mycoviruses Using RNA Sequencing and Electron Microscopy

The sequencing of dsRNA extracted from strain 266 MV revealed the presence of five viral RNAs, constituting the genomes of four distinct mycoviruses. The MiSeq run generated a total of 4,090,354 reads. After quality trimming, 3,692,466 reads were left. Normalized reads were mapped against *Fusarium graminearum* PH-1 genome/transcriptome accessions (chromosomes 1 to 4; 13,401 predicted transcripts and mitochondrion sequences; BioProject PRJNA243 and NC_009493), and the remaining 17,641 reads were de novo assembled into contigs. Complete genome sequences were determined using RACE and were deposited in NCBI GenBank (Table 1). The BLAST alignment against the virus reference databases revealed five hits with various degrees of identity to genomes of mycoviruses (Table 1). Two sequences were assigned to RNA 1 and RNA 2 of *Fusarium poae virus* 1 (FpV1), with a high percentage of identity to the nuclear core sequences from the primary host species. Icosahedral particles about 35–40 nm in diameter, visible using electron microscopy of mycelial extracts, confirmed the infection with FpV1 (Figure 3A). This suggests the horizontal transmission of the mycovirus between *F. poae* and *F. graminearum*. The genomic segments of FpV1 were not further characterized, given their high similarity to reference sequences.

Two sequences showed an up to 90% identity with a Soybean leaf-associated negative-stranded RNA virus and Cryphonectria parasitica ambivirus 1, and they were tentatively assigned to Fusarium graminearum negative-stranded RNA virus 2 (FgSV2) and Fusarium graminearum ambivirus 1 (FgAV1), according to the guidelines of the International Committee on Taxonomy of Viruses (ICTV). A sequence of 2606 nt showed a 96% identity with Plasmopara viticola lesion-associated mitovirus 13. The genomes of these three mycoviruses have been characterized in detail and are presented in Section 2.3.

RT-PCR with virus-specific primers (Section 4.2, Appendix A Table A1) detected all five segments in RNA extracts from strain 266 MV, yet failed to detect them in strain 266.

### 2.3. Characterization of Mycoviral Genomes

#### 2.3.1. *Fusarium graminearum ambivirus* 1 (FgAV1)

The genome of *Fusarium graminearum ambivirus* 1 (FgAV1), with a length of 4579 nt, exhibited a circular ssRNA structure with two open reading frames in opposite directions: ORF A in the 5′–3′ orientation and ORF B in the 3′–5′ orientation (Figure 4), as is a typical feature of ambiviruses. ORF A encodes a protein of 709 amino acids (aas), predicted to be the RNA-dependent RNA polymerase (RdRp), while ORF B codes for a protein of 662 aas, with unknown function. FgAV1 shares low sequence similarity with the sequences in public databases, with highest similarity (73%) to Cryphonectria parasitica ambivirus 1 (MT354566). The genome structure and phylogenetic analysis of the putative RdRp protein are shown in Figure 4.

The replication of ambiviruses has only recently been unraveled. In their seminal paper, Forgia et al. [30] showed that ambiviral genomes replicate via a rolling-circle mechanism, which has so far mainly been known from much smaller replicons of viroids. In contrast to viroids, the replication of ambiviruses is catalyzed by their own RdRp. Concatenated genomes produced by rolling-circle replication are dissected by the action of two antiparallel hammerhead ribozymes. The predicted secondary structures of the ribozymes of FgAV1 and their cleavage sites, kindly provided by Dr. Beatriz Navarro and Dr. Francesco Di Serio (Istituto per la Protezione Sostenibile delle Piante, CNR, Bari, Italy) are shown in Appendix A Figure A2.

**Figure 4 toxins-16-00131-f004:**
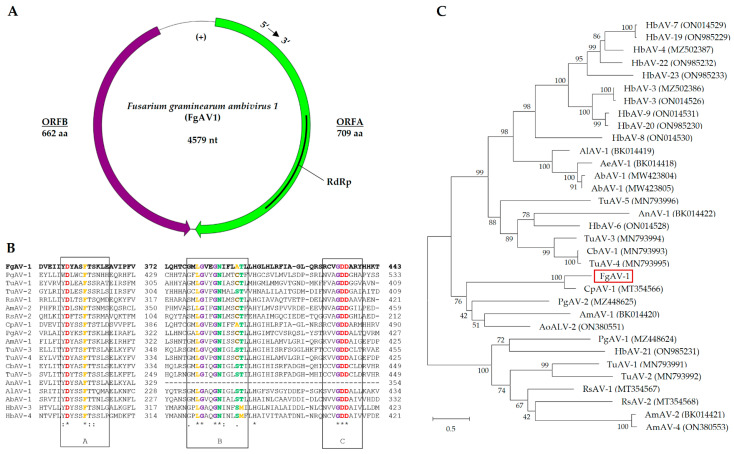
Genome organization of *Fusarium graminearum ambivirus* 1 (FgAV1). (**A**) Schematic representation of the circular genome. Open reading frames A and B are represented by purple and green colors, respectively. (**B**) Alignment of RdRp sequences; conserved motifs, according to [31], are framed in black boxes. Amino acid sequences were selected from the accessions used for the generation of the dendrogram in part C (Appendix A Table A2). The alignment was created with Clustal Omega [32]. Asterisks (*) mark positions with fully conserved amino acid residues, colons (:) mark residues with highly similar physicochemical properties that score > 0.5 in PAM 250 [33], and periods (.) mark residues scoring 0.0 to 0.5 in PAM 250. (**C**) Dendrogram of putative RdRp sequence of FgAV1, other ambiviruses, and ambivirus-like sequences. The dendrogram was constructed using the Neighbor-Joining method [34] implemented in MEGA X 10.1.8 [35]. Bootstrap values in % (1000 replicas) are shown. The tree is drawn to scale, with branch lengths corresponding to evolutionary distances, which were estimated using a JTT matrix-based method [36]. The scale is in substitutions per site.

#### 2.3.2. *Fusarium graminearum negative-stranded RNA virus* 2 (FgNSRV2)

The genome sequence assigned to *Fusarium graminearum negative-stranded RNA virus* 2 is 8939 nt long and contains five putative ORFs (Figure 5). The 3′- and 5′- UTRs are 127 and 5 nt long, respectively. The fourth and largest ORF encodes a putative protein of 1941 aas with an RdRp domain (pfam00946), an mRNA-capping region (pfam14318), and a methyltransferase domain (cl27811). Phylogenetic analysis of ORF 4 (Figure 5) placed FgNSRV2 in the order *Mononegavirales*, family *Mymonaviridae*, and genus *Sclerotimonavirus*. The ORF 1 of FgNSRV2 encodes a 274 aa long protein that contains three conserved regions, which were putatively identified as a helix-rich domain (TIGR04523), an ATPase (COG1196), and a phosphatidylglycerophosphatase domain (PRK03918) according to the NCBI Conserved Domain Database (CDD). Comparative sequence analysis with BLASTp showed that ORF 1 shared high homology (>75%) with proteins encoded by *Soybean leaf-associated negative-stranded RNA virus* 1 (ALM62222) and *Fusarium graminearum negative-stranded RNA virus* 1 (UNG44333), which belong to the family *Mymonaviridae*. No conserved domain was found in ORF 2 (388 aa), ORF 3 (56 aa), nor ORF 5 (195 aa). However, ORF2 exhibits a high level of sequence similarity to other members of the order *Mymonaviruses* from *Fusarium* spp., including ATP75709 of *F. graminearum*, UNG44330 of *F. asiaticum*, and UWK02084 of *F. proliferatum*.

The sequence of FgNSRV2 contains a conserved gene-junction section, which is commonly found and regarded as a characteristic feature in Mononegaviruses [38,39]. An alignment of the gene-junction sequence of FgNSRV2 with sequences from other known Mononegaviruses is presented in Figure 6. So far, only a few exceptions with six rather than five genome ORFs were reported in Mononegaviruses [39]. One gene-junction section per ORF was reported for SsNSRV-1, whereas four junctions were found in FgNSRV1 and FgNSRV2.

#### 2.3.3. *Fusarium graminearum mitovirus* 2 (FgMV2)

The genome of *Fusarium graminearum mitovirus* 2 (FgMV2) has a length of 2634 nt (Figure 7). The 5′-UTR and 3′-UTR sequences are 194 nt and 169 nt long. A single ORF, which encodes a protein of 757 aas (L-Protein), was found using the mitochondrial genetic code. In the same way as described for a mycovirus of *Sclerotinia sclerotiorum* [41], FgMV2 can be assigned to Class II mitoviruses. The genome of *Plasmopara viticola lesion associated mitovirus* 13 (MN539774) is the most similar to FgMV2, sharing 96.3% similarity (Appendix A Table A2). The investigation of secondary structures with RNAfold version 2.6.3 [42] revealed stem-loop structures, characteristic of mitoviruses [43], which are believed to protect ssRNA genomes from enzymatic degradation (Appendix A Figure A3).

### 2.4. Transfection of Mycoviruses from Strain 266 MV into Mycovirus-Free Strain and the Effect of Mycoviruses on Radial Growth and Sporulation

Transfection of mycoviruses from strain 266 MV to strain 668 was carried out via the co-incubation of the strains on agar plates (Appendix A Figure A1). The selection of transfectants was facilitated by a resistance marker carried by the acceptor strain (Section 4.4) Two transfected strains were verified by the presence of viral dsRNA fragments in agarose electrophoreses. In the case of FpV1, the transmission was also verified by the visualization of viral particles using electron microscopy (Figure 3B).

Strain 266 MV harbored four mycoviruses (Table 1). To find out which mycoviruses have been transfected into strain 668 MV and also which mycoviruses were still present in the cured strain 266, we tested all strains used in this study for the presence of each mycovirus using RT-PCR (Section 4.2) with mycovirus-specific primers (Table A1 and Table 2). Differences among the strains in mycoviral load allowed us to correlate phenotypic features with mycoviral infection.

The radial growth of the strain was measured at 25 °C (Section 4.5). Strain 266 MV exhibited an irregular growth pattern with altered pigmentation (Figure 2B), but no effect of mycoviral infection on the radial growth rate was observed (Figure 8A). In contrast, the radial growth rate of strain 668 MV was reduced by about 28% as compared to strain 668. Colonies of the transfected strain 668 MV were deeply red, similar to the colonies of strain 266 MV (Figure 2A,B).

The capacity of all four strains to produce conidia in liquid SNA media varied to a large extent, as can be seen in Figure 8B,D, though parallel cultures were grown under identical conditions. No differences in the sporulation between strains 266 and 266 MV, nor between 668 and 668 MV were observed.

### 2.5. Effect of Mycoviral Infection on the Production of Secondary Metabolites

#### 2.5.1. Production of Trichothecenes and Aurofusarin

To investigate the effect of infection with mycoviruses on the capability of *F. graminearum* to produce trichothecenes and aurofusarin, strains 266, 266 MV, 668, and 668 MV were cultivated on rice kernels. Nine replicates of strains 266 and 266 MV and eight replicates of strains 668 and 668 MV were processed. Concentration of aurofusarin, DON, 3ADON, and 15ADON were determined using HPLC–MS/MS (Section 4.6), and fungal DNA was quantified using qPCR (Section 4.3). The concentrations of secondary metabolites were then normalized to fungal DNA to compensate for the differences in fungal biomass accumulated by strains infected with mycoviruses and mycovirus-free strains.

In five out of six combinations of strain pairs and trichothecenes, lower amounts of trichothecenes were found in cultures infected with mycoviruses (Figure 9). The suppression of trichothecene synthesis was substantial and highly statistically significant, except for 15ADON in strain pair 668/668 MV.

The accumulation of aurofusarin was suppressed in strain 668 MV as compared to 668, but it remained unchanged in strain 266 MV as compared to 266. Of note, cultures of strain 668 accumulated more aurofusarin (2.2 to 4.5 g/kg) than cultures of strain 266 (1.0 to 3.5 g/kg). This can likely be explained by the disruption of zearalenone synthesis in strain 668 (Section 4.1), which increased the pool of acetyl-CoA available for the synthesis of aurofusarin.

#### 2.5.2. Effect of Mycoviruses on the Production of Volatile Compounds

Volatile organic compounds (VOCs) emitted by fungal cultures on PDA were analyzed using GC–MS (Section 4.7). Only metabolites detected in both genotypes were quantified. In both pairs of strains, infection with mycoviruses strongly stimulated the emission of two isomeric methyl-butanols, and it strongly suppressed the emission of the trichothecene precursor trichodiene (Figure 10). The identity of trichodiene was confirmed by comparing the retention index (RI) and fragmentation pattern with an authentic standard (Appendix A Figure A4).

### 2.6. Effect of Mycoviral Infection on Food Preference in Fungivorous Arthropods

To assess whether the infection of *F. graminearum* with mycovirus affected the feeding behavior of fungivores, we carried out a dual-choice food preference assay with the collembolan *Folsomia candida*. In strain 266, at all times, more springtails fed on cultures infected with mycoviruses than on uninfected cultures (Figure 11A). In strain 668, springtails did not distinguish between infected and uninfected mycelia in the first half of the experiment, but more animals fed on mycelia infected with mycoviruses in the second half of the experiment (Figure 11B).

The statistical significance of the springtails’ preference for infected mycelia was estimated using cumulative binomial probabilities (Section 4.8). The preference of springtails for strain 266 MV over 266 was statistically significant (*p* ≤ 0.03) at all time points, except at 6 h and 8 h. Over the entire experiment, the preference of *F. candida* for 266 MV over 266 was highly significant with *p* < 0.0001. In contrast, the preference of springtails for strain 668 MV over 668 was not significant (*p* > 0.05) at any single time point; however, over the duration of the entire experiment, the springtails preferred feeding on strain 668 MV over 668 with a *p* value of 0.028.

## 3. Discussion

### 3.1. Generation of Isogenic Strains Differing in the Presence of Mycoviruses

Pairs of isogenic fungal strains differing in the presence of mycoviruses are pivotal in studies investigating the effect of mycoviruses on fungal phenotype. Such strains can be obtained by curing strains harboring mycoviruses or transmission into mycovirus-free strains. Our attempt to cure mycoviruses of strain 266 MV by ribavirin and cycloheximide failed (results not shown) and, indeed, varying success rates of this procedure have been reported [44]. An attempt to cure mycoviruses by mitotic segregation was more successful, resulting in the successful curing of two mycoviruses (Table 2). A second pair of isogenic strains differing in the load of mycoviruses was constructed by transmitting mycoviruses from strain 266 MV to 668 (Table 2). We have not succeeded in obtaining *F. graminearum* strains harboring single mycoviruses, as pioneered by Lee et al. [25]. Still, comparison of the effect of mycoviruses on fungal phenotype in the two isogenic strains allowed for the tentative assignment of phenotypic effects to infection with specific mycoviruses. Assignment to single-virus isolates could be inferred by comparison of the combinations reported in Table 2; however, caution should be taken due to the potential different response of the *Fusarium* genotypes to the same virus and/or to virus interactions.

A cautionary remark concerns the pair 688/688 MV. Strains 688 and 688 MV may not be strictly isogenic, because the formation of heterokaryons and parasexual recombination might have taken place during the anastomosis. The same limitation holds for the results obtained with strains harboring single mycoviruses [25]. These strains were obtained via protoplast fusion, which, too, may cause the formation of heterokaryons, followed by parasexual recombination. Strictly speaking, rigorous verification would require multiple pairs of strains obtained independently, regardless of whether the strains were obtained by anastomosis or protoplast fusion. This situation is comparable to the use of gene disruption mutants in the elucidation of gene function. Furthermore, any infection is modulated by the host genotype. Major effects observed in this study were similar in both genotypes used (Section 3.4, Section 3.5 and Section 3.6), but some effects were observed only in strain 688, which may be due to different sets of mycoviruses carried by the strains or due to differences between the genotypes of the strains. While keeping these restrictions in mind, the comparative analysis of fungal strains harboring different sets of mycoviruses is a powerful strategy to untangle the effects of mycoviruses on the fungal phenotype.

### 3.2. Effect of Mycoviral Infection on Fungal Growth and Sporulation

Colonies of strains 266 and 266 MV differed in growth pattern and pigmentation (Figure 2A,B), but they exhibited the same radial growth rate (Figure 8A). Colonies of 668 MV and 668 looked similar in the first passage. We propose that this behavior can be explained by a low virus titer in freshly transfected cultures, as reported for the transfection of FgV-ch9 [22]. After several passages, colonies of 668 MV turned dark red and their margin became irregular, similarly to 266 MV. The radial growth of 668 MV was strongly reduced (Figure 8C). From these results and the comparison among the strains (Table 2), we could infer that the reduction in the growth rates of strain 668 MV was caused by mycovirus FpV1. An alternative explanation, which holds for all inferences of this kind in the following sections, is that the genotypes 266 and 668 respond differently to the presence of the same mycovirus. If this is the case, the growth reduction in strain 668 MV may have been caused by mycovirus FgV1. In any case, neither FgNSRV1 nor FgMV2 were the cause of the growth rate reduction in strain 668 MV, as they were not transmitted to the 668 wild-type strain.

Reduced sporulation [45,46] and deformation of conidia [47] due to infection of *F. graminearum* with mycoviruses has been reported previously. Visual inspection of cultures on agar indicated that conidia production may have been reduced in 266 MV and 668 MV, but the effect was not statistically supported.

### 3.3. Effect of Mycoviral Infection on the Accumulation of Aurofusarin

The observation of the stronger pigmentation of strains carrying mycoviruses (e.g., cf. Figure 2A,B) suggested that mycoviral infection stimulated the synthesis of red pigments. The red pigment aurofusarin is the key defense compound of *F. graminearum* protecting the fungus from predators [12]. Although the colonies of strains infected with mycoviruses appeared darker than uninfected strains, aurofusarin levels in infected cultures did not increase. In strain 668 MV, the accumulation of aurofusarin was actually reduced, and in strain 266 MV it remained unchanged (Figure 9D,H). The comparison among the strains (Table 2) suggests that the mycovirus responsible for the reduction in aurofusarin synthesis in 668 MV was FpV1.

Could other pigments be responsible for the stronger coloration of mycovirus-infected colonies? Red pigments of *F. graminearum,* other than aurofusarin, are rubrofusarin and carotenoids. Rubrofusarin is a precursor of aurofusarin; its overproduction would, therefore, cause an increase in the production of aurofusarin. Carotenoids do not appear to contribute substantially to the red pigmentation of *F. graminearum*, because strains without aurofusarin form whitish colonies with a faint yellow–orange tone [48]. Dominant carotenoids of *F. graminearum* are orange neurosporaxanthin and rose-red torulene [49]. We cannot exclude that the increased pigmentation of 266 MV and 688 MV was caused by an overproduction of torulene, but we think that it is unlikely. Examination of fungal colonies in this work (Figure 2A,B) and published photos (Figure 2B in [22], Figure 2A in [45], and Figure 3A in [47]) suggests another explanation. Uninfected colonies form abundant aerial mycelia and are fluffy, while colonies of infected strains are slimy with few aerial mycelia. We believe that this explains why colonies with mycoviruses appear darker.

### 3.4. Effect of Mycoviral Infection on the Production of Trichothecenes in F. graminearum

The accumulation of DON and both acetylated derivatives of DON in strain 266 was strongly suppressed by mycoviral infection (Figure 9). In strain 668, the accumulation of DON and 3ADON but not 15ADON was suppressed by mycoviral infection. Suppression of the synthesis of trichothecenes in *F. graminearum* was reported in 2002 for the first time [21]; in the abstract, the authors wrote that fungal strains with a mycovirus (which was later named FgV1) exhibited “*decreased (60-fold) production of trichothecene mycotoxins*”. The difference was 16- rather than 60-fold but, more importantly, DON concentrations were not normalized to fungal biomass. Thus, lower DON concentrations may have resulted solely from a lower biomass. The error remained unnoticed for two decades, and it is still perpetuated today. For instance, a recent review [50] claims that viruses substantially reduced trichothecene production, citing another review [51] instead of a primary source, and [51] cites [21] without inspecting its content. These examples show how important it is for review authors to examine the primary literature rather than relying on abstracts and secondary sources. Twelve years after their discovery of the mycovirus FgV1, the authors showed that it, indeed, suppresses trichothecene synthesis in *F. graminearum* [25]. Using protoplast fusion, they generated four strains of *F. graminearum*, each harboring a single mycovirus, and showed that three other mycoviruses did not substantially affect trichothecene synthesis (though they observed a slight increase in normalized trichothecene levels in one of these cultures that was statistically significant).

In our work, suppression of trichothecene synthesis by mycoviruses was observed in both pairs, 266/266 MV and 688/688 MV. The only mycovirus that was absent in both 266 and 688 and present in both 266 MV and 688 MV was FgAV1. Therefore, the most likely cause of the suppression of trichothecene synthesis was the ambivirus FgAV1. FgAV1 is a circular ssRNA virus with a genome length of 4.6 kb (Section 2.3.1), while FgV1 is a dsRNA virus with a genome length of 7.5 kbp [25]. Thus, different mycoviruses are apparently able to suppress trichothecene synthesis in *F. graminearum*.

### 3.5. Effect of Mycoviral Infection on the Emission of Volatile Metabolites

Mycoviruses suppressed the emission of sesquiterpene trichodiene in both fungal genotypes (Figure 9), which was expected because trichodiene is a precursor of trichothecenes [2,4].

In both genotypes, mycoviral infection strongly stimulated the emission of 2-methyl-1-butanol and 3-methyl-1-butanol (Figure 10). Considering the distribution of mycoviruses among the strains (Table 2), an analogous reasoning regarding the cause of the suppression of trichothecene synthesis (Section 3.4) suggests that the most plausible cause of the increased production of 2- and 3-methyl-1-butanol was infection with the ambivirus FgAV1.

### 3.6. Effect of Mycoviruses on the Attractiveness of Fungal Mycelia for Collembolans and Its Possible Causes

Collembolans preferred strain 266 MV over 266 at all time points (Figure 11), and the preference was highly significant over the entire experiment (Section 2.5). Instead, the number of collembolans feeding on 668 MV exceeded those on 668 only in the second half of the experiment. At both 5 h and 6 h, the proportion of animals feeding on 668 MV was high, yet not statistically significant, due to a low number of animals. Over the entire experiment, the preference for strain 668 MV over 668 was significant (Section 2.5).

These observations give rise to the following question: can these effects be accounted for by the modulation of the synthesis of secondary metabolites? In strain 668, mycoviruses suppressed the accumulation of aurofusarin (Figure 9H), which is the major (and possibly the only, see [12]) metabolite protecting *F. graminearum* from fungivory. Thus, suppression of aurofusarin synthesis likely contributed to the preference of collembolans for strain 668 MV over 668. It cannot be the only reason, however, because aurofusarin levels in 266 and 266 MV were comparable (Figure 9D).

Has the reduction in DON content contributed to the enhanced attractiveness of mycovirus-infected strains for collembolans? A large body of literature has been devoted to investigating the feeding preferences of collembolans, but few have addressed the role of trichothecenes directly. In most of these studies, grains colonized with *Fusarium* were used as a food source, which obscured the cause of observed effects (e.g., [52,53]). We are aware of only two reports on the effect of DON on collembolans under controlled conditions. B. Ulber et al. (unpublished results quoted in [54]) found that collembolans fed with high concentrations of DON did not show any symptoms, and the disruption of trichothecene synthesis in *F. graminearum* has not affected the food preference of *F. candida* (Suppl. Figure 5 in [12]). Both reports disproved the hypothesis that a reduced content of trichothecenes B accounted for the increased attractiveness of infected strains to collembolans. Recurring exposure of collembolans to trichothecenes B might have led to resistance. Selection for resistance to aurofusarin might have been prevented by vast amounts accumulating in mycelia, which denied partially resistant mutants of a fitness advantage [12].

Fungal VOCs are known to trigger olfactory responses in collembolans and modulate their food preference [55]. Our results revealed two groups of VOCs able to explain the effect of mycoviruses on the attractiveness of *F. graminearum* for collembolans. The first of them is trichodiene. We excluded the role of trichothecenes B produced by *F. graminearum*, but trichodiene is also a precursor of trichothecenes A and macrocyclic trichothecenes, which are substantially more toxic than trichothecenes B. It is conceivable that the ability of collembolans to avoid mycelia that emit trichodiene was selected by the toxicity of highly toxic trichothecenes produced by fungi other than *Fusarium* spp. The hypothesis could be easily tested by an olfactometric experiment.

The second candidate for the cause of the increased attractiveness of infected mycelia for collembolans is isomeric methylbutanols (Figure 10). Low concentrations of 2-methyl-1-butanol saturated the antennal receptors of *Orchesella cincta* and *Tomocerus flavescens* [56] and attracted collembolan *Onychiurus armatus* [57]. 3-methyl-1-butanol has not been tested in these studies, though the production of this VOC by the fungi investigated in these studies was explicitly mentioned; we assume that the compound was not commercially available at that time. The hypothesis that 2-methyl-1-butanol and/or 3-methyl-1-butanol contributed to the attractiveness of mycovirus-infected strains to *F. candida* can be easily tested in an olfactometric experiment.

Does the stimulation of VOC emission by mycoviruses possess an ecological function? Plant viruses modulate the attractiveness of their hosts to animal vectors, increasing their chance of propagation [58,59]. The dissemination of mycoviruses by fungivores has been demonstrated in a single case [16]. We suggest that the hypothesis that the stimulation of the emission of volatile attractants contributes to the dissemination of mycoviruses can be tested by monitoring the spread of mycoviruses in soil microcosmoses exposed to different levels of fungivory.

## 4. Materials and Methods

### 4.1. Fungal Strains and Cultures

The strains of *Fusarium graminearum* used are listed in Table 3. *F. graminearum* 266 MV, deposited in the DSMZ culture collection (Braunschweig, Germany) as *F. graminearum* DSM 116490, was isolated in 2018 from a naturally infected maize ear grown in Pocking (Bavaria, Germany), with a disease rating of 5% according to a modified symptom scale [60], as previously described [61]. *F. graminearum* 668 (DSM 113709), kindly provided by Wilhelm Schäfer (University of Hamburg, Hamburg, Germany), is a zearalenone-deficient mutant (△ZEN) generated from wildtype strain *F. graminearum* 1003 by disrupting the polyketide synthase gene *PKS4* [62]. The strain carries a hygromycin B (*HPH*) resistance gene, which served as a resistance marker in transmission experiments. Strain 266 (DSM 116489) is a descendant of 266 MV (DSM 116490) that spontaneously lost mycoviruses (for details, see Section 3.1). Strain 668 MV (DSM 116488) is a transfectant of 668 (DSM 113709), harboring mycoviruses from 266 MV (DSM 116490). All strains were grown on potato dextrose agar (PDA, Carl Roth, Karlsruhe, Germany) at 25 °C in the dark.

### 4.2. RNA Extraction, Sequencing, RACE, RT-PCR, and Phylogenetic Analysis

Mycelium was scrubbed from the surface of a PDA plate, snap-frozen in liquid nitrogen, and ground in a ceramic mortar. Total RNA was extracted from frozen mycelium using TRIzol (Invitrogen, Carlsbad, CA, USA), according to the manufacturer’s instructions. The quality of RNA extracts was assessed using electrophoresis in nondenaturing 2.2% agarose gels [63].

The cDNA was synthetized with random octamer primers and Maxima H Minus Reverse Transcriptase (Thermo Fischer Scientific, Waltham, MA, USA), followed by second-strand synthesis using the NEB Next Ultra II Non-Directional RNA Second Strand Module (New England Biolabs, Ipswich, MA, USA). Following on from column purification with a NucleoSpin Gel and PCR Clean-up kit (Macherey-Nagel, Dueren, Germany), an Illumina Nextera XT DNA library was prepared and quantified according to the manufacturer’s recommendations (Qubit, Thermo Fischer Scientific), and the size of DNA fragments was assessed with a 2100 Bioanalyzer system and a High Sensitivity DNA Assay Kit (Agilent, Darmstadt, Germany). After normalization, the samples were sequenced on an Illumina MiSeq instrument as paired-end reads (2 × 300 bp). Contigs were de novo assembled and further analyzed in GeneiousPrime (Biomatters, Auckland, New Zealand) for virus discovery by BLAST alignment and taxonomic assignment.

Complete genome sequences of the identified viruses were obtained using RACE, according to [64]. Both 5′ and 3′ RACEs (rapid amplification of cDNA ends) were performed for FpV1, FgSV1, and FgMV2. For this purpose, cDNA was synthesized using a virus-specific primer, tailed with A, C, G, or T, and the products were amplified using a virus-specific primer and a primer matching the respective tail. Specifically, 4 µL of dsRNA solution, 1 µL of the respective primer (10 μM; Eurofins Genomics, Hamburg, Germany), and 7 µL of H_2_O were incubated at 95 °C for 3 min and then immediately transferred to liquid nitrogen. A pre-warmed (50 °C) RT mix consisting of 4 µL 5× RT buffer, 1 µL dNTPs (10 mM each; Carl Roth, Karlsruhe, Germany), 0.25 µL Maxima H Minus Reverse Transcriptase, and 0.5 µL Ribolock RNase Inhibitor, all from Thermo Fisher Scientific (Darmstadt, Germany). In total, 4.25 µL H_2_O was added and the reaction was incubated for 60 min at 50 °C, for 15 min at 55 °C, for 15 min at 60 °C, and for 5 min at 85 °C. After purification using the SureClean Plus system (Bioline, Luckenwalde, Germany), cDNA was tailed with A, C, G, and T. To this end, 3 µL of purified cDNA were incubated with 4 µL of 5X reaction buffer, 1 µL terminal deoxynucleotidyl transferase (Thermo Fisher Scientific, Darmstadt, Germany), 1 µL of the respective dNTP (100 mM), and 11 µL of H_2_O for 30 min at 37 °C, followed by 10 min at 70 °C. The subsequent PCR reaction, consisted of 3 µL tailed cDNA, 15 µL of 2X Phusion Flash High-Fidelity PCR Master Mix (Thermo Fisher Scientific, Darmstadt, Germany), 1.5 µL of virus- and tail-specific primer (10 μM each), and 9 µL water. The PCR program started at 98 °C for 15 s, followed by 34 cycles of 98 °C for 5 s, primer annealing (55 to 59 °C) for 5 s, 72 °C for 15 s, and ended with a final elongation of 5 min. Half of the sample was loaded onto a 1% agarose gel, purified using a NucleoSpin Gel and PCR Clean-up Kit (Macherey-Nagel, Düren, Germany) in 20 µL final volume, following manufacturer’s instructions, and either sequenced directly or cloned into a standard cloning vector (CloneJET PCR Cloning Kit, Thermo Fisher Scientific, Darmstadt, Germany). If a single fragment was clearly visible on the gel, the second half of the sample that was not loaded onto the agarose gel was purified using the SureClean Plus system (Bioline, Luckenwalde, Germany) in 20 µL final volume and sequenced directly.

For the reverse transcription of specific fragments of mycoviral genomes, 2 µL of total RNA were digested with 2 U/µL DNAseI (New England Biolabs, Frankfurt am Main, Germany) in DNase reaction buffer (10 mM Tris-HCl, 2.5 mM MgCl_2_, 0.5 mM CaCl_2_, pH 7.6). The quality of dsRNA was assessed using agarose gel electrophoresis (Section 2.2). RNA was reverse transcribed using primers specific for viral sequences (Appendix A Table A1) as follows: 3 µL of dsRNA extract were incubated together with 1 µL of the appropriate sense and antisense primer (10 μM each; Eurofins Genomics, Hamburg, Germany) for 5 min at 95 °C and then immediately cooled on ice. This is followed by the addition of 4 µL 5X RT buffer (Thermo Fisher Scientific), 0.5 µL dNTPs (10 mM each; Carl Roth), 1 µL RevertAid Reverse Transcriptase (20 U/µL) (Thermo Fisher Scientific), and 9.5 µL H_2_O and an incubation of 45 min at 42 °C. For the subsequent PCR, a reaction mix consisting of 2 µL cDNA, 5 µL 2X Phusion Flash High-Fidelity PCR Master Mix (Thermo Fisher Scientific, Darmstadt, Germany), 1 µL of sense and antisense primer each (10 μM each; salt-free; Eurofins Genomics, Hamburg, Germany), and 1 µL H_2_O, started at 98 °C for 15 s, followed by 34 cycles of 98 °C for 5 s, primer annealing (55 to 59 °C) for 5 s, 72 °C for 15 s, and ended with a final elongation of 5 min. Subsequently, the sample was loaded onto a 1% agarose gel and selected fragments were purified using a Macherey-Nagel NucleoSpin Gel and PCR Clean-up Kit (Macherey-Nagel, Düren, Germany) and were sequenced.

Following translation into amino acids, the complete gene for RNA-dependent RNA polymerase (RdRp) was used to determine the evolutionary relationships and genetic distances to reference genomes in the NCBI database and construct phylogenetic dendrograms [65,66]. Sequence alignments were created and Neighbor-Joining dendrograms [34] were constructed with MEGA 10.1.8 [35].

### 4.3. DNA Extraction, DNase Treatment, and Quantification of Fungal Biomass

In rice cultures, fungal DNA was used as a proxy for biomass. Mycelia harvested from a liquid culture (for DNA standards) and entire rice cultures consisting of rice grains and fungal mycelium (samples) were freeze-dried and ground to a fine powder in a reciprocal mill (MM400, Retsch, Haan, Germany). DNA was extracted using a cetyltrimethylammonium-based protocol [67]. To differentiate between DNA and RNA bands in agarose electrophoresis, nucleic acid extracts were treated with 2 U/µL DNaAse I (New England Biolabs, Frankfurt am Main, Germany) in DNase reaction buffer (10 mM Tris-HCl, 2.5 mM MgCl2, 0.5 mM CaCl2, pH 7.6) for 1 h at 37 °C.

Fungal biomass in grains was determined using quantitative PCR (qPCR), as described [67], using two replicates of densely spaced standards (see Figure A2 in [60]). The biomass of fungal colonies growing on agar was determined directly, as described in Section 4.7.

### 4.4. Transmission of Mycoviruses between F. graminearum Strains

Transmission of viral RNAs from *F. graminearum* 266 MV to *F. graminearum* 668 was achieved by hyphal anastomosis during co-cultivation according to Darissa et al. [22]. Briefly, an agar block of 6 mm diameter overgrown with mycelium of strain 266 MV was placed on a PDA plate and incubated for 5 days at 25 °C in the dark at room temperature. A second agar block with virus-free hygromycin-resistant strain 668 was positioned at a distance of 1 cm. After 7 days of incubation in the dark, several agar blocks were cut out from the area where the fungal colonies met, transferred to PDA plates containing 50 µg/mL of hygromycin B1 (Carl Roth GmbH, Karlsruhe, Germany), and incubated for 5 days. Single spores were harvested by rinsing the cultures with sterile tap water and plating dilutions of the spore suspensions onto agar plates. This workflow is visualized in Figure A1.

Single-spore isolates were assigned to the 266 or 668 genotype using PCR with the primer pair FGPKS4f1 and FGPKS4r1 (Appendix A Table A1), which encloses the area of *PKS4* that was disrupted in the acceptor strain 668. PCR was carried out using the conditions used for the amplification of *TEF-*1*α* [29]. The presence of viral dsRNAs in the acceptor strain was also detected by agarose electrophoresis and confirmed using reverse transcription PCR (see Section 4.2).

### 4.5. Determination of Radial Growth Rate and Sporulation

Radial growth rate was determined on potato dextrose agar (PDA) plates (Carl Roth, Karlsruhe, Germany). Round agar blocks covered with mycelium with a diameter of 5 mm were placed onto agar plates and incubated in the dark at 25 °C for 48 h. Eight plates were used for each estimate. From each plate, four measurements of the distance from the agar block to the tips of mycelial hyphae were taken at a 90° angle.

In order to assess the effect of infection with mycovirus on sporulation, infected and uninfected strains were cultivated in 100 mL of nutrient-deprived broth (SNA), containing 9.9 mM KNO_3_, 7.3 mM KH_2_PO_4_, 0.3 mM MgS0_4_ × 7H_2_O, 6.7 mM KCl, 1.1 mM glucose, and 0.6 mM sucrose [68] for 7 days at 25 °C with shaking at 130 rpm. Spores were collected from culture supernatant via centrifugation at 4500 rcf for 10 min, suspended in 1 mL of sterile tap water, and counted in a Thoma chamber. Macroconidia were counted with a Thoma chamber. Spore concentrations below the limit of quantification (5 × 10^4^ spores/mL) were replaced with limit of quantification, analogously to left-censored data in analytical chemistry.

### 4.6. Quantification of Trichothecenes and Aurofusarin

Rice cultures were prepared as previously described [69] and were grown for 21 d at 25 °C. The extraction and HPLC–MS/MS analysis of trichothecenes were previously described [69]. Aurofusarin was extracted separately, using 1 g of freeze-dried and ground culture material and 10 mL methanol, and analyzed using HPLC–MS/MS, essentially as described by Xu at al. [12]. MS/MS transitions, limits of detection (LODs), and limits of quantification (LOQs) are listed in Appendix A Table A1. To compensate for differences in fungal biomass, concentrations of metabolites were normalized to fungal DNA determined using qPCR (Section 4.3).

### 4.7. Analysis of Volatile Compounds

Petri dishes (35 mm diameter; Sarstedt AG & Co. KG, Nuembrecht, Germany) were filled with 4 mL of PDA, inoculated with *F. graminearum* strains, and cultivated at 25 °C in the dark. After seven days, the lid was removed, and the plate was transferred into a top-open headspace glass vessel (50 mm diameter, 115 mm height) with an inlet and an outlet for airflow at opposite sides (openings of 9 mm diameter). The inlets were closed with screw caps, and the upper opening was sealed with polyester oven bag foil (Confresco Frischhalteprodukte GmbH & Co. KG, Minden, Germany). The glass vessel was then further incubated for 24 h at 25 °C in darkness. After incubation, the vessels were attached to an air inflow adjusted to 320 mL purified air/min and the air was sucked out at a rate of 300 mL/min with a Laboport vacuum pump (KNF Neuberger GmbH, Freiburg, Germany). Before entering the headspace, the air was filtered through two hydrocarbon/moisture traps (Agilent Technologies, Model HT200-4). Headspace sampling was carried out for 2 h in darkness at 24 °C and 45% relative humidity. Volatiles were collected with Porapak Q filters (VCT-1/4-3-POR-Q, www.ars-fla.com (accessed on 8 June 2023) placed in the outflow opening and were eluted with 150 µL dichloromethane. Porapak filters were rinsed with 1 mL dichloromethane before each sampling. Both empty Petri dishes and Petri dishes filled with PDA were analyzed to identify background volatiles. Six replicates of each culture were analyzed.

200 ng of tetralin standard (20 ng/µL; Sigma-Aldrich Chemie GmbH, Taufkirchen, Germany) was added to each sample and 40 µL of the spiked sample was transferred to a new glass vial with a glass insert for GC–MS analysis, which was carried out on an Agilent Technologies 7890 GC System with Agilent 5977B GC/MSD. An HP5-MS analytical column was used (30 m × 0.25 mm inner diameter, 0.25 μm film thickness, Agilent Technologies, Santa Clara, CA, USA). In total, 2 µL of the eluate were injected in splitless mode at 220 °C and 18.84 psi. The oven temperature was held at 40 °C for 3 min, raised to 320 °C at a rate of 8 °C min^−1^, and held at this temperature for 8 min. Helium was used as a carrier gas at a constant flux of 1.5 mL min^−1^.

Chromatograms were analyzed using Enhanced ChemStation, MSD ChemStation F.01.03.2357 (Agilent, Darmstadt, Germany). VOC identities were confirmed by comparing mass spectra and retention indices (RIs) with the values listed by NIST (National Institute of Standards and Technology, USA, www.webbook.nist.gov (accessed on 14 April 2023). The identity of trichodiene was confirmed by the comparison of retention index and fragmentation spectrum with the authentic standard (+)-trichodiene (HPC Standards, Cunnersdorf, Germany). Only signals occurring in all strains were quantified. For relative quantification, peak areas were normalized with the area of the internal standard tetralin.

Intensities of signals that were found in both *F. graminearum* genotypes were normalized to fungal biomass as follows: Fungal mycelium was separated from agar by placing agar overgrown with mycelia into a beaker with water and heating the mixture in a microwave until agar melted and dissolved in water. The mycelium was then separated by filtration, freeze-dried, and weighed. Volatile levels were normalized to the mycelial weight. The significance of differences between means was assessed using Student’s *t*-test or the Wilcoxon rank sum test. Samples that did not emit any detectable VOCs were excluded from the analysis.

### 4.8. Food Preference Assay

The springtail *Folsomia candida* was reared in Petri dishes on a layer of plaster of Paris and activated charcoal (13:1, *w*/*w*) at 20 °C in darkness. Collembolans were fed weekly with dry baker’s yeast (Lucullus Food Service GmbH & Co. KG, Trittau, Germany) ad libitum. The plate was aerated during feeding. Tap water was added when needed to maintain humidity levels. Before being used in experiments, collembolans were transferred to a new plate with pure plaster of Paris and starved for 48 h.

Food preference assays with collembolans were performed in plastic Petri dishes (9 cm diameter; Sarstedt AG & Co. KG, Nuembrecht, Germany) on plaster. All trials were carried out with one animal per plate to avoid crowding effects in three repetitions with ten replicates each. The animals were placed in the middle of the arena and had to choose between the mycelium of infected and healthy strains of *F. graminearum*, presented as agar plugs (9 mm diameter) overgrown with fungal mycelium. The results were evaluated by counting the animals that fed on infected or healthy mycelium or that have been located in the periphery (‘no choice’). Counting was repeated once per hour for 8 h. During the experiment, the animals were kept in the dark at 21 °C.

The *p*-value was estimated as a cumulative binomial probability that the same or a larger number of animals would feed on cultures with mycoviruses under the conditions of the experiment if the animals had chosen the food source randomly.

### 4.9. Transmission Electron Microscopy

Fresh mycelium was scrubbed from the surface of a PDA plate of both the infected and uninfected cultures, snap-frozen in liquid nitrogen, and ground in a ceramic mortar. Mycelium powder was resuspended in 1 mL of a 0.1 M sodium-phosphate buffer (2% PVP, 0.2% sodium sulfite, 0.05% sodium azide) and incubated for 30 min at 4 °C in a rotary shaker at 400 rpm (Thermomixer comfort; Eppendorf SE, Hamburg, Germany). Then, 800 µL of chloroform/isoamyl alcohol (24:1, *v*/*v*) were added to the mixture and the samples were centrifuged at 10,000× *g* for 10 min (Centrifuge 5415D; Eppendorf SE, Hamburg, Germany). Then, 600 µL of the supernatant were transferred to a new 2 mL tube with 800 µL chloroform/isoamylalkohol (24:1, *v*/*v*) and the centrifugation step was repeated. After that, 600 µL of the supernatant was transferred to a new 1.5 mL Eppendorf tube, and 200 µL of 30% (*w*/*v*) PEG 6000 and 100 µL 5 M NaCl were added. The samples were mixed by inverting the tube 7 times. Precipitation was performed for 1 h at 4 °C in a rotary shaker at 400 rpm. Afterwards, the sample was centrifuged at 10,000× *g* and 4 °C for 15 min. The supernatant was discarded, and the pellet was dissolved in 20 µL of 0.1 M sodium-phosphate buffer. All samples were stored at 4–7 °C until transmission electron microscopy analysis.

Electron microscopy was performed at the facilities of the institute for Epidemiology and Pathogen Diagnostics of the Julius Kühn-Institute in Braunschweig. Briefly, a copper grid (G400EM-C3, 400 lines/inch square mesh; Plano, Wetzlar, Germany) was coated with a 0.5% Pioloform in chloroform solution and sputtered with carbon (Mikrotechnik Dr. Hert, München, Germany). Mycelium extract was applied on the surface of Parafilm, and a coated grid was placed on the top of the extract. Virions were allowed to adsorb on the grid for 5 min, the grid was washed with distilled water, drained with filter paper, and stained with a 1% uranyl acetate solution (Riedel de Haën, Seelze, Germany). Transmission electron microscopy was performed with a Tecnai G2 Spirit TWIN microscope (Hillsboro, ON, USA) system. Photos were taken with a CCD camera, model WA-Veleta (Olympus Life Science Solutions, Hamburg, Germany).

## Figures and Tables

**Figure 1 toxins-16-00131-f001:**
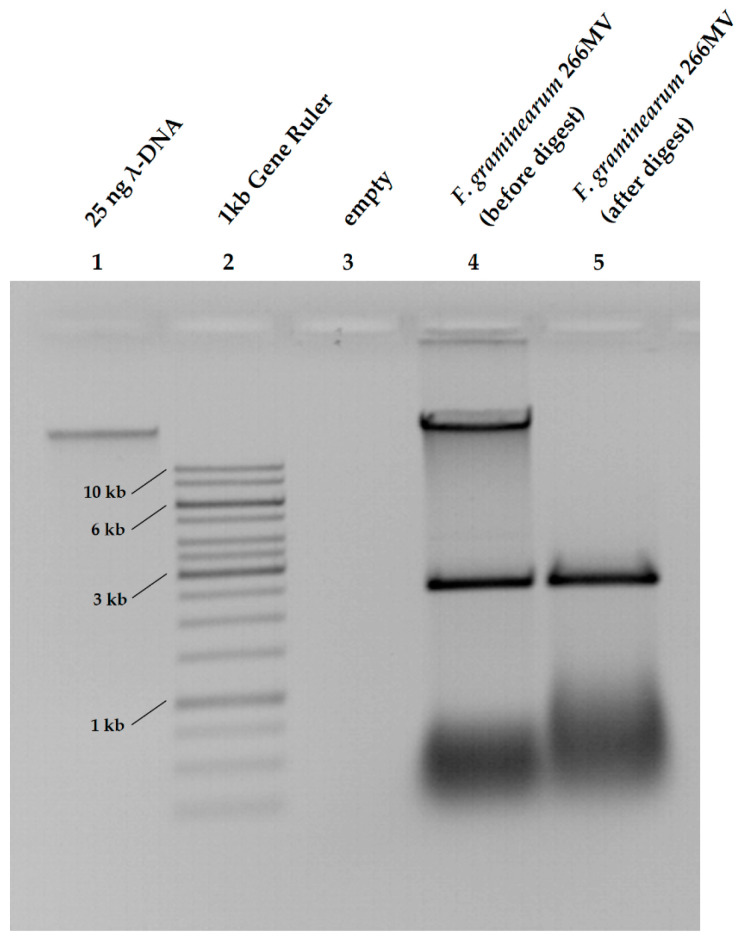
DNase digestion of nucleic acids extracted from mycelia of *F. graminearum* 266 MV.

**Figure 2 toxins-16-00131-f002:**
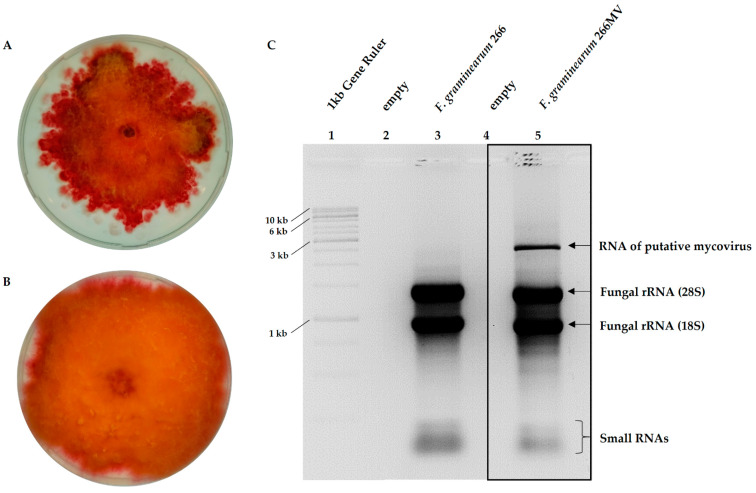
Comparison of *F. graminearum* 266 MV and 266, differing in the presence of an extrachromosomal RNA element. Left: growth pattern of 266 MV (**A**) and 266 (**B**) on PDA, 14 d after inoculation. Right (**C**): agarose gel electrophoresis of total RNA extracted from strains 266 and 266 MV.

**Figure 3 toxins-16-00131-f003:**
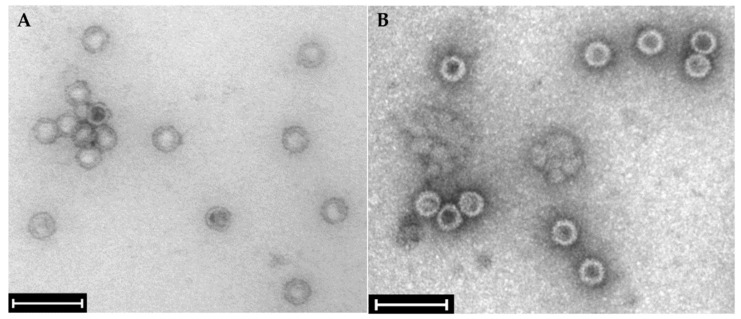
Transmission electron micrograph of virus-like particles purified from the mycelium of *Fusarium graminearum*-carrying mycoviruses. (**A**) Wild type strain 266 MV; (**B**) strain 668 MV, into which mycoviruses have been transfected (Section 3.4). Isometric particles of approximately 38 nm resemble Betapartitiviruses, such as *Fusarium poae virus* 1 (FpV1). The bar represents 100 nm.

**Figure 5 toxins-16-00131-f005:**
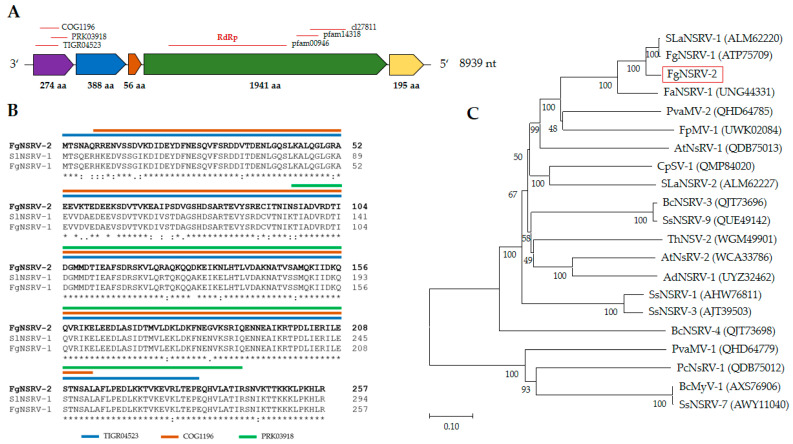
Genome organization of *Fusarium graminearum* negative-stranded RNA virus 2 (FgNSRV2). (**A**) Schematic representation of the genome of FgNSRV2, including five open reading frames (ORFs). Conserved functional domains and accession numbers, identified in BLAST/CDD analysis, are indicated above the ORFs [37]. (**B**) Alignment of the sequence of ORF1 from FgNSRV2 with highly similar sequences from SlNSRV1 (ALM62220) and FgNSRV1 (ATP75709) from NCBI. Full species names and accession Nos. are shown in Appendix A Table A2. (**C**) Dendrogram constructed from the sequences of ORF4, putatively encoding RdRp, and similar sequences obtained from NCBI. For the construction of the alignment and dendrogram, symbols for amino acid similarity, bootstrap values, and the scale refer to Figure 4.

**Figure 6 toxins-16-00131-f006:**
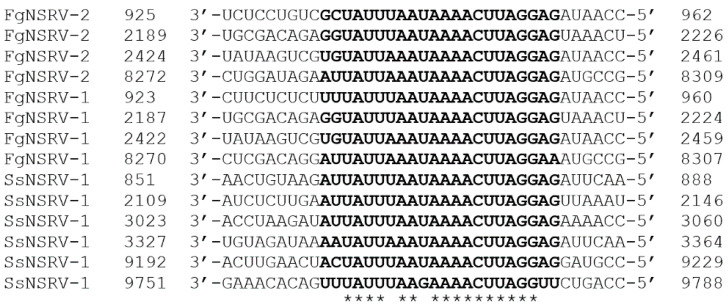
Alignment of the gene-junction sequences of FgNSRV2 with SsNSRV-1 (KJ186782, [39]) and FgNSRV-1 [40]. Nucleotide sequences of ORFs are labelled according to Figure 5. Numbers indicate the position of the first and last nucleotide of the motif in the genome. The conserved gene-junction sequences (A/U/G)(G/U/A/C)UAUU(U/A)AA(U/G)AAAACUUAGG(A/U)(G/U) are highlighted in bold. Asterisks * mark positions with fully conserved residues.

**Figure 7 toxins-16-00131-f007:**
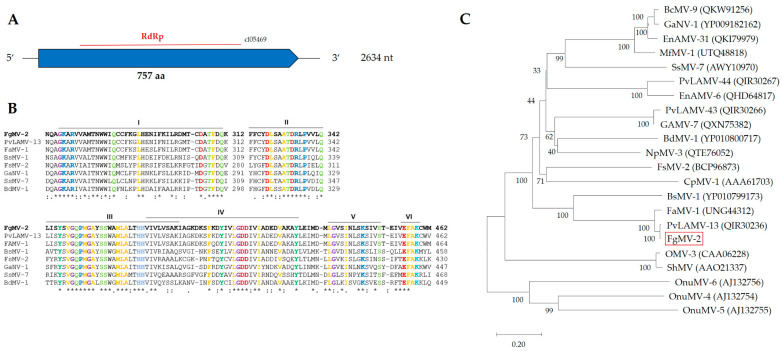
Genome of *Fusarium graminearum mitovirus* 2 (FgMV2). (**A**) Single ORF encoding RdRp, (**B**) alignment of conserved motifs, and (**C**) dendrogram of RdRp sequences. For the explanation of symbols and description of the software, see Figure 4.

**Figure 8 toxins-16-00131-f008:**
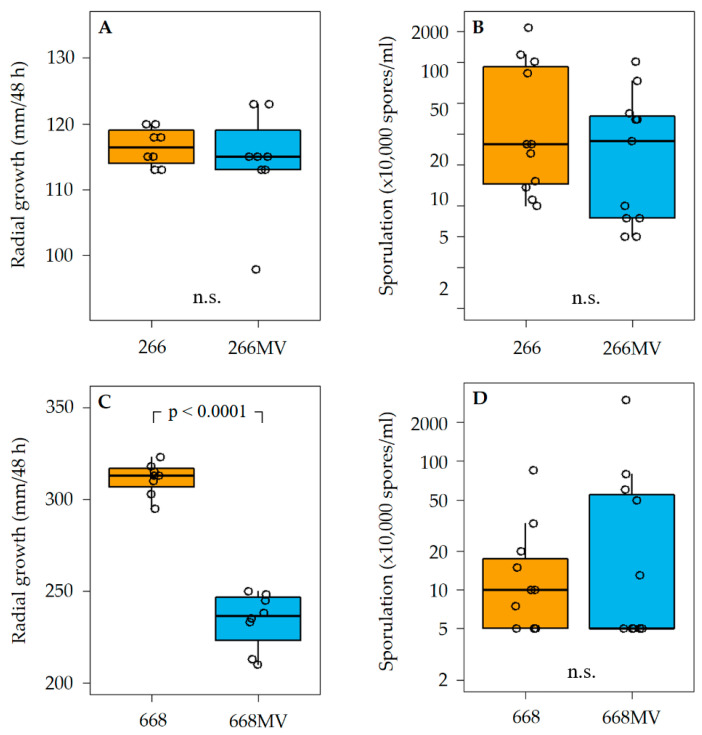
Effect of mycoviral infection on radial growth and sporulation capacity of *F. graminarum*. In each panel, results obtained with two isogenic strains differing merely in the presence of mycoviruses are shown. Panels (**A**,**B**) show the effects of mycoviral infection on growth (**A**) and sporulation (**B**) in strain 266. Panels (**C**,**D**) show the effects of mycoviral infection on growth (**C**) and sporulation (**D**) in strain 668. Radial growth rates were estimated on PDA after 48 h of incubation at 25 °C in the dark. Sporulation capacity was estimated by growing cultures in liquid media for 7 d at 25 °C (Section 4.5). n.s. stands for *p* > 0.1.

**Figure 9 toxins-16-00131-f009:**
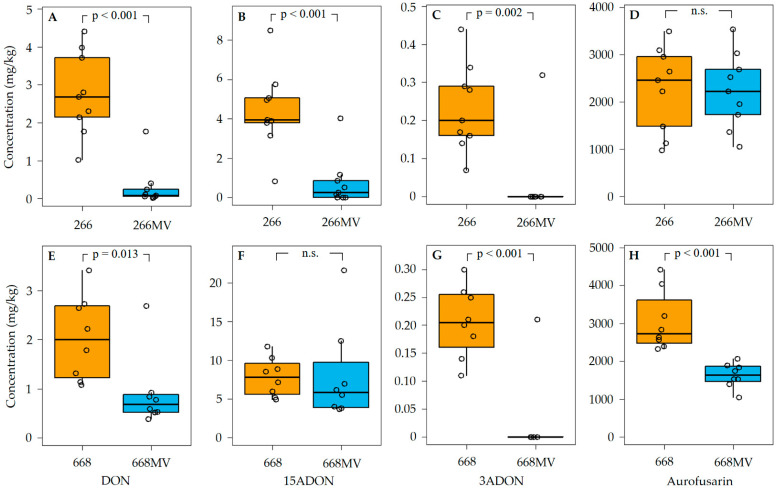
Effect of infection with mycoviruses on the production of trichothecenes and aurofusarin by *F. graminearum*. Concentrations are expressed in mg/kg dry weight of rice cultures normalized to fungal biomass. The effect of mycoviruses on the production of DON (panels **A**,**E**), 15ADON (**B**,**F**), 3ADON (**C**,**G**), and aurofusarin (**D**,**H**) by strains 266 (**A**–**D**) and 668 (**E**–**H**) is shown. For details regarding culture conditions, metabolite extraction, and HPLC–MS/MS analysis, see Section 4.6.

**Figure 10 toxins-16-00131-f010:**
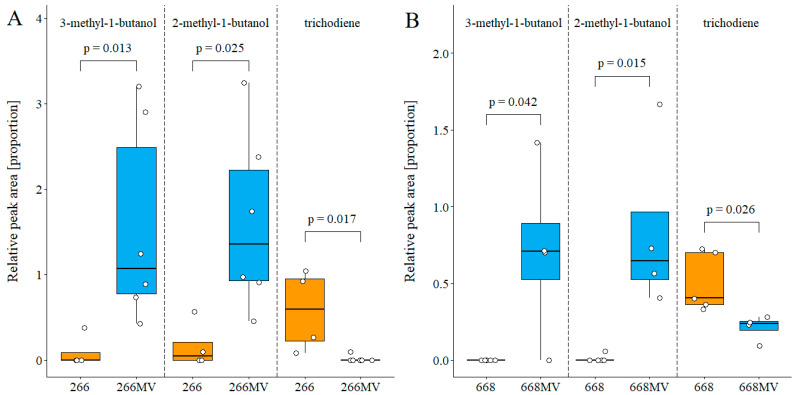
Effect of mycoviral infection on the emission of volatile metabolites by *Fusarium graminearum.* (**A**) Comparison of strains 266 and 266 MV. (**B**) Comparison of strains 668 and 668 MV. Sample collection and analysis is described in Section 4.7. Statistical comparisons between treatments were performed with Student’s *t*-test or the Wilcoxon ranked test, according to data structure. Boxplot whiskers extend to the most extended data point within the 1.5 interquartile range. The upper box limit represents the third quartile, the lower box limit represents the first quartile, and the box line represents the second quartile.

**Figure 11 toxins-16-00131-f011:**
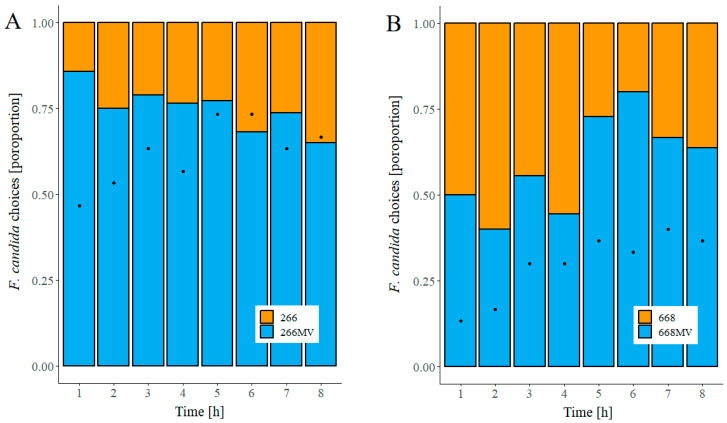
Effect of mycoviral infection on food preference of *Folsomia candida*. (**A**) Strains 266 and 266 MV and (**B**) strains 668 and 668 MV. Food choice was determined as the proportion of responding individuals that fed on the mycelium of a particular strain. Response, represented by black dots, was defined as the proportion of individuals feeding on mycelial plugs.

**Table 1 toxins-16-00131-t001:** Mycovirus sequences assembled from *F. graminearum* 266 MV.

Name	Size [nt]	Accession No.	Most Similar Genomes Using NCBI BLASTn			
			Closest Reference Sequence	Accession No.	Genome [nt]	Similarity [%]
FpV1	2298	ON969095	*Fusarium poae virus* 1 RNA 1	LC150606	2431	98.6
	2205	ON969096	*Fusarium poae virus* 1 RNA 2	OK524181	2212	98.8
FgNSRV2	8939	ON969097	Soybean leaf-associated negative-stranded RNA virus	NC075296	9041	98.0
FgMV2	2634	ON969098	Plasmopara viticola lesion associated mitovirus 13	MN539774	2607	96.3
FgAV1	4579	ON969099	Cryphonectria parasitica ambivirus 1	MT354566	4579	73.0

**Table 2 toxins-16-00131-t002:** Presence of mycoviruses in *F. graminearum* strains.

Strain	FpV1	FgNSRV1	FgMV2	FgAV1
266	yes	no	yes	no
266 MV	yes	yes	yes	yes
668	no	no	no	no
668 MV	yes	no	no	yes

**Table 3 toxins-16-00131-t003:** Strains of *Fusarium graminearum*.

Strain	Short Label	Properties
*F. graminearum* DSM 116490 *	266 MV	Natural isolate harboring mycoviruses
*F. graminearum* DSM 116489	266	Descendant of 266 MV that lost two mycoviruses
*F. graminearum* DSM 113709	668	Hygromycin-resistant strain without mycoviruses
*F. graminearum* DSM 116488	668 MV	Transfectant of 668 harboring two mycoviruses from 266 MV

* DSMZ-German Collection of Microorganisms and Cell Cultures GmbH, Inhoffenstraße 7 B, 38124 Braunschweig, Germany.

## Data Availability

Data are provided in the paper and its appendices. Nucleotide sequences have been deposited in NCBI, Acc. Nos. are listed in Appendix A Table A2. All strains are available from the DSMZ-German Collection of Microorganisms and Cell Cultures GmbH, Inhoffenstraße 7 B, 38124 Braunschweig, Germany.

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
