# Peer review of "Mycoviruses Increase the Attractiveness of *Fusarium graminearum* for Fungivores and Suppress Production of the Mycotoxin Deoxynivalenol"

_toxins, 2024, doi:10.3390/toxins16030131_

Round 1
Reviewer 1 Report
Comments and Suggestions for Authors
It’s an attractive alternative to employ viruses to combat fugal crop pathogens. In this study, the authors identified a new species of virus based on its genome and investigated the effects of these RNA viruses on the phenotype, toxins production and the predator’s preference in F. gramineaum. Their work is novel and meaningful, and the manuscript is also organized well. It could be considered for the acceptance to be published in our journal after the following questions are addressed.
1.The lane 5 in Figure 2C was the profiles of RNA from the virus-infected F. gramineaum, and the labels of Lane 4&5 in Figure 1 were the profiles of RNA from virus. However, their labels were the same, which would be misleading.
2.Considering the host specificity of viruses, it is important to shed light on if there are the same effects of these RNA viruses on other F. gramineaum strains.
3.The results of in vivo experiments in plant leaves could be more persuasive regarding the biological effects of these RNA viruses on F. gramineaum.
Reviewer 2 Report
Comments and Suggestions for Authors
The authors have done excellent job on this manuscript. While there are no major concerns in the quality and presentation of the manuscript, one issue does highlight. The issue being extensive criticism of article 21 in citation list on the error or 16 vs 60-fold. While it can be confirmed that the original article (21) has error based on the data from that paper, I find the extensive criticism in this manuscript beyond requirement.
I suggest authors to revise sections/text on this aspect on lines 52-58, 416-428;
